# Exogenous Copper Application for the Elemental Defense of Rice Plants against Rice Leaffolder (*Cnaphalocrocis medinalis*)

**DOI:** 10.3390/plants11091104

**Published:** 2022-04-19

**Authors:** Boon Huat Cheah, Wen-Po Chuang, Jing-Chi Lo, Yi Li, Chih-Yun Cheng, Zhi-Wei Yang, Chung-Ta Liao, Ya-Fen Lin

**Affiliations:** 1Department of Agronomy, National Taiwan University, Taipei 10617, Taiwan; boonhuatcheah@ntu.edu.tw (B.H.C.); wenpo@ntu.edu.tw (W.-P.C.); freebike01@gmail.com (Y.L.); 2Department of Horticulture and Biotechnology, Chinese Culture University, Taipei 11114, Taiwan; ljq13@ulive.pccu.edu.tw; 3Crop Improvement Division, Taoyuan District Agricultural Research and Extension Station, Taoyuan City 32745, Taiwan; kurama630@tydais.gov.tw (C.-Y.C.); zwyang@tydais.gov.tw (Z.-W.Y.); 4Crop Environment Division, Taichung District Agricultural Research and Extension Station, Changhua County 51544, Taiwan; liaoct@tdais.gov.tw

**Keywords:** micronutrient supplement, trace elements, *Oryza sativa* L., rice tolerance, insect pest

## Abstract

Metals that accumulate in plants may confer protection against herbivorous insects, a phenomenon known as elemental defense. However, this strategy has not been widely explored in important crops such as rice (*Oryza sativa* L.), where it could help to reduce the use of chemical pesticides. Here, we investigated the potential of copper (Cu) and iron (Fe) micronutrient supplements for the protection of rice against a major insect pest, the rice leaffolder (*Cnaphalocrocis medinalis*). We found that intermediate levels of Cu (20 μM CuSO_4_) and high concentrations of Fe (742 μM Fe) did not inhibit the growth of *C. medinalis* larvae but did inhibit rice root growth and reduce grain yield at the reproductive stage. In contrast, high levels of Cu (80 μM CuSO_4_) inhibited *C. medinalis* larval growth and pupal development but also adversely affected rice growth at the vegetative stage. Interestingly, treatment with 10 μM CuSO_4_ had no adverse effects on rice growth or yield components at the reproductive stage. These data suggest that pest management based on the application of Cu may be possible, which would be achieved by a higher effective pesticide dose to prevent or minimize its phytotoxicity effects in plants.

## 1. Introduction

The rice leaffolder (*Cnaphalocrocis medinalis* Guenée; Lepidoptera: Pyralidae) is a major insect pest of rice (*Oryza sativa* L.) in Asia [1]. In Taiwan, severe infestation usually occurs during the second cropping season from June to October, which is characterized by hot and humid weather that favors the growth and reproduction of this pest, leading to yield losses of 18–24% [2,3]. In the paddy field, *C. medinalis* larvae instinctively build shelters or feeding chambers by folding a leaf longitudinally using silk strands that attach to the leaf margins [4]. The larvae feed by scraping mesophyll tissue from within the folded leaf, resulting in longitudinal white and transparent streaks [5]. A single larva can damage a number of rice leaves, with cumulative effects that reduce photosynthesis and thus cause yield losses [6,7]. The main strategy to prevent *C. medinalis* outbreaks in rice fields is the application of topical insecticides, but this causes health issues, as well as environmental and ecological damage [8,9].

Integrated pest management is a holistic solution that includes the use of resistant cultivars, biological control and good field management practices, such as the effective use of fertilizers [10]. The macronutrient content of fertilizers is prioritized over the micronutrient content [11], but optimal micronutrients not only promote plant growth but may also enhance biotic stress tolerance [12,13]. For example, copper (Cu)-deficient plants are more susceptible to pathogens [12], and desert locusts (*Schistocerca gregaria*) prefer the leaves of *Noccaea caerulescens* plants deficient in zinc (Zn) compared to plants grown in the presence of intermediate or high concentrations of Zn [14].

The elemental defense hypothesis (also known as the inorganic defense hypothesis) proposes that plants accumulate high levels of inorganic elements as a defensive strategy to protect themselves from pests and pathogens [15]. Inorganic elements may contribute to plant defense directly (by defensive enhancement) or indirectly (by joint effects) [16,17]. In the defensive enhancement hypothesis, metal ions accumulating in plants confer protection once they reach a threshold concentration that is toxic toward pests [16]. For example, artificial diet experiments indicated that plants containing 20–300 mg/kg dry weight (DW) of cobalt (Co) or 140–1000 mg/kg DW of nickel (Ni), both within the so-called accumulator range, as well as 200–400 mg/kg DW of Zn (within the normal physiological range), may directly inhibit the growth and development of beet armyworm (*Spodoptera exigua*) due to metal toxicity [18]. On the other hand, the joint effects hypothesis proposes that metal ions and organic defense chemicals have additive or synergistic effects against herbivores and pathogens [17,19]. The mechanisms of elemental defense differ according to which elements, plant species and pests are involved [20].

Elemental defense has been investigated mainly in dicotyledonous plants and/or heavy metal hyperaccumulators, focusing on arsenic (As), cadmium (Cd), Ni, selenium (Se) and Zn, but rarely on Cu and iron (Fe) [21,22]. Few studies have considered the feasibility of elemental defense in major crops such as rice, although silicon (Si) fortification has been shown to protect rice against *C. medinalis* larvae by promoting cell silicification, reducing the soluble protein content, and inducing the biosynthesis of defensive enzymes and metabolites such as jasmonic acid (JA) [9,22,23]. As a rule of thumb, an element that confers protection by elemental defense in plants should reach a certain range of endogenous concentrations that can suppress the growth of pests or pathogens without negative effects on plant growth [24]. We examined whether the elemental defense of a rice cultivar susceptible to *C. medinalis* (Taoyuan No. 3) can be induced by exogenous micronutrient supplements, focusing on Cu and Fe. We evaluated the effects of different concentrations of Cu and Fe on *C. medinalis*, rice vegetative growth and rice reproductive growth in an attempt to identify the ideal concentration for elemental defense.

## 2. Results

### 2.1. Cu Impedes the Growth and Development of C. medinalis Larvae

To test the elemental defense potential of Cu and Fe, we compared the growth of *C. medinalis* larvae reared on a susceptible rice cultivar (Taoyuan No. 3) grown in substrates containing specific Cu or Fe supplements (Figure 1). The fresh weight (FW) of *C. medinalis* larvae feeding on rice plants supplied with 80 μM Cu was 17% lower than the control group (0.05 μM Cu) 3 days post-infestation (dpi), but not after 6 days (Figure 1A). The 80 μM Cu treatment decreased the relative growth rate (RGR) of larvae after 3 days (0.38 compared to 0.42 in the control group; Figure 1B), hence fifth-instar larvae were predominant (92%) in the 80 μM Cu treatment group after 6 days in contrast to the predominant prepupae (60%) in the control group (Figure 1C). After 9 days, 38% (5/13) of the larvae in the 80 μM Cu treatment group showed delayed development, remaining in the prepupal stage, while all larvae in the control group had entered the pupal stage (Figure 1D).

In contrast to the 80 μM Cu treatment group, we observed no effects on larval FW or RGR in response to either 65 or 265 mg/L Fe (equivalent to 197 and 742 μM Fe, respectively) after 3 and 6 days, compared to the control group (15 mg/L Fe, equivalent to 61 μM Fe) (Figure 1E,F). Similarly, the Fe treatments did not delay *C. medinalis* development compared to the control group after 6 and 9 days (Figure 1G,H).

These results showed that Fe has no potential for elemental defense against the *C. medinalis*, at least under the conditions we tested, whereas Cu fulfilled the first criterion of elemental defense by delaying *C. medinalis* growth and development above a certain threshold concentration.

### 2.2. Vegetative Rice Growth Is Inhibited by Cu in a Dose-Dependent Manner but Is Not Inhibited by Fe

Having identified the threshold Cu concentration that inhibits pest growth and development, we next investigated the range compatible with vegetative growth in rice. We therefore grew rice plants in the presence of 20 and 80 μM Cu and examined the effect 15 days after treatment (DAT) (Figure 2). The 80 μM Cu treatment clearly inhibited vegetative shoot growth at 15 DAT compared to the 0.05 μM Cu control group (Figure 2A). We examined several morphological and molecular parameters, and found that the 80 μM Cu treatment reduced the shoot height, relative chlorophyll content and biomass of leaf blades, sheaths and roots compared to the control group (Figure 2B,C and Appendix A). An intermediate treatment (20 μM Cu) caused milder physiological changes at 15 DAT, particularly a reduction in root FW compared to the control group (Figure 2B,C and Appendix A).

To ensure that the observed physiological changes were caused by Cu treatment, we measured the micronutrient concentrations of the rice plants. The Cu content of the roots in the 20 μM Cu treatment group was 11.5-fold higher than in the control group (590 vs. 51 mg/kg DW), increasing to 21.7-fold higher (1113 vs. 51 mg/kg DW) in the 80 μM Cu treatment group (Figure 2D). However, the Cu concentration in the shoots increased by a similar amount in both treatment groups, from 23 mg/kg DW in the control group to 58 mg/kg DW in the 20 μM Cu treatment group (2.5-fold higher) and 64 mg/kg DW in the 80 μM treatment group (2.8-fold higher) (Figure 2E). Elemental analysis showed that the concentrations of Mn and Fe were unaffected by either Cu treatment, whereas the shoot Zn concentration was reduced by the 80 μM Cu treatment (Appendix A).

The analysis of defense-related phytohormones showed that the concentrations of JA and its bioactive derivative JA-isoleucine in rice leaves increased in the 20 μM Cu treatment group but not in the 80 μM Cu treatment group at 15 DAT (Figure 3A,B). The concentration of salicylic acid (SA) in the rice sheath increased in the 80 μM Cu treatment group (Figure 3C).

We also tested the effects of 65 and 265 mg/L Fe on the vegetative growth of rice (Appendix A). Although shoot growth at 15 DAT appeared to be similar in the control group and both treatment groups (Appendix A), quantitative analysis revealed some changes in response to higher concentrations of Fe (Appendix A). The plants in both treatment groups were taller (8.1% and 3.5% in the 65 and 265 mg/L Fe groups, respectively), the relative chlorophyll content increased (by 8.3% and 13.1%, respectively), and several tissues accumulated more biomass, particularly the leaf blade DW (+55%) and sheath DW (+121%) in the 265 mg/L Fe treatment group (Appendix A).

The Fe treatments did not influence intracellular Fe levels in a predictable manner (Appendix A). For example, the 65 mg/L Fe treatment had no effect on the Fe content of the roots at 15 DAT, whereas the Fe content of the shoots increased 28.5-fold from 53 mg/kg DW in the control group to 1517 mg/kg DW after treatment (Appendix A). This suggests that Fe taken up by rice roots was rapidly and efficiently translocated to the shoots (Appendix A). In contrast, the 265 mg/L Fe treatment affected the Fe content of neither the roots nor the shoots at 15 DAT, suggesting that a heavy metal exclusion mechanism had been triggered to prevent metal toxicity (Appendix A). Elemental analysis showed that Mn levels were unchanged, Cu levels in the shoot decreased in the 265 mg/L Fe treatment group, and Zn levels in the root increased in both Fe treatment groups (Appendix A). The concentration of defense-related hormones was unaffected by the Fe treatments (Appendix A).

In summary, both Cu treatments inhibited the vegetative growth of rice plants with varying levels of severity, whereas both Fe treatments promoted the vegetative growth of rice plants instead. It is therefore clear that vegetative rice plants are more susceptible to metal toxicity caused by Cu than Fe.

### 2.3. The Growth and Yield Components of Reproductive Rice Plants Are Unaffected by 10 μM Cu

Given the severe effects of 80 μM Cu on vegetative rice plants, we evaluated the effect of lower concentrations (10, 20 and 30 μM Cu) for 120 days compared to the 0.05 μM Cu control (Figure 4, Table 1). Morphologically, the three Cu treatments did not affect the growth of aboveground tissues (Figure 4A) but root growth was inhibited by the 20 and 30 μM Cu treatments in a dose-dependent manner (Figure 4B). The treatments had no effect on the shoot height and relative chlorophyll content of rice at the reproductive stage (Figure 4C,D). None of the treatments significantly affected the grain yield, but the 30 μM Cu treatment delayed booting by 7–8 days, presumably due to the inhibition of root growth (Table 1, Figure 4A,B).

Notably, the aboveground effect of 265 mg/L Fe on shoots was opposite to that on roots at the reproductive stage (Appendix A). Shoot height was not affected by either Fe treatment, but the relative chlorophyll content of plants exposed to 265 mg/L Fe increased by 27.5% compared to the control group (Appendix A). Conversely, root growth was severely inhibited in the presence of 265 mg/L Fe (Appendix A) and this probably explained the 29.5% lower grain yield compared to control plants (Table 1).

Taken together, root growth and yield components in reproductive rice were negatively affected by exposure to 265 mg/L Fe but not 10 μM Cu. Based on the initial results for the effect of Cu on *C. medinalis* development, effective pest management would require a treatment option that achieved a dose of 80 μM Cu on the leaf surface, to inhibit feeding by the insect larvae, while ensuring that the concentration in plants remained within 10~30 mg Cu/kg DW.

## 3. Discussion

In an ideal form of elemental defense, the concentration of metal ions in plants is sufficient to inhibit pest growth and development without negative effects on the plant [24]. Elemental defense studies have mainly focused on heavy metal hyperaccumulators and the levels of As, Cd, Ni, Se and Zn, whereas we examined the potential of Cu and Fe to achieve protection against the *C. medinalis* based on the effects of each element on the growth of *C. medinalis* and a susceptible rice cultivar, Taoyuan No. 3 (Figure 5).

We found that *C. medinalis* larvae feeding on rice plants treated with 80 μM Cu grew and developed more slowly than the control group and did not gain as much FW (Figure 1A–D). Similarly, maize (*Zea mays*) plants exposed to 80 μM Cu inhibited fall armyworm (*Spodoptera frugiperda*) growth due to the priming of herbivore-induced JA and volatile organic compounds in maize leaves by heavy metal stress [25]. Another elemental defense study showed that pepper plants (*Capsicum annuum* L.) exposed to 50 μM Cu were able to tolerate verticillium wilt better than controls, which was attributed to a Cu-induced defense response resulting in the induction of defensive genes that increased the availability of peroxidases and phenolic compounds [26]. The foliar application of Cu(OH)_2_ fungicide elicits an SA-dependent defense mechanism in *Arabidopsis thaliana* that governs the effectiveness of the fungicide against *Peronospora parasitica* [27].

Cu, therefore, has the potential to protect plants against insect pests and pathogens but only if the effective concentration is compatible with normal plant growth. However, we found that the exogenous application of 80 μM Cu increased the Cu concentration in the shoots to 64 mg/kg DW, severely inhibiting vegetative growth at 15 DAT (Figure 2B–E and Appendix A). The optimal Cu concentration in rice shoots is ~10 mg/kg DW, and anything above ~30 mg/kg DW is harmful, inducing toxicity symptoms such as the loss of chlorophyll, the inhibition of photosynthesis, metabolic disruption and ultimately, stunted growth and low yields [28,29,30]. Furthermore, we also observed the severe inhibition of root growth at the reproductive stage in the presence of 30 μM Cu, which delayed booting and heading (Figure 4A,B). In agreement with our results, a previous pot soil experiment showed that root growth was more severely affected than shoot growth in rice plants at the reproductive stage when the Cu concentration in the soil was 50–150 mg/kg, whereas both shoot and root growth were severely inhibited at concentrations of 300–1000 mg/kg [31]. Rice grain yields decreased in relation to the soil Cu concentration, with ~10% yield losses at 100 mg/kg, ~50% yield losses at 300 mg/kg and up to 90% yield losses at 1000 mg/kg [31]. The minimum lethal (530 mg/kg DW) and sublethal (140 mg/kg DW) concentrations of Cu against beet armyworm neonates were much higher than the aforementioned normal range of Cu concentrations in rice shoots, whereas the equivalent values for *C. medinalis* larvae remain unknown [19].

Our results indicate that Cu is not ideal as the basis of elemental defense against the *C. medinalis*, given the absence of a concentration that is both toxic toward the pest and harmless toward the plant. Any gains achieved by the suppression of pest development in the presence of 80 μM Cu would be offset by its negative effects on rice plants at the vegetative and reproductive stages (Figure 5). One potential solution is the use of nanotechnology to increase the effective dose of Cu on the plant surface while preventing the uptake of excess Cu and the resulting inhibition of plant growth, which could be achieved by the exogenous topical application of Cu nanoparticles. For example, a field experiment comparing the antifungal activity of foliar applied Cu-based nanoparticles and other commercial agrochemicals on *Phytophthora infestans* infected tomato (*Solanum lycopersicum*) and showed that the nanoparticles exhibited higher activity than the commercial agrochemical at a low concentration without causing any deleterious effect on plants [32]. Another study revealed that the Cu-based nanoparticles inhibited the growth of *Xanthomonas axonopodis* pv. *punicae*, a pathogen causing bacterial blight in pomegranate, at 0.2 ppm, i.e., >10,000 times lower than that suggested for Cu-oxychloride fungicide [33]. Recently, an adhesive nanopesticide was reported to show better long-term control efficacies against *C. medinalis* (Guenée) and *Chilo suppressalis* (Walker) than the commercial Benevia insecticide and also had no apparent effect on the growth of rice [34]. Nevertheless, it is noteworthy that there is a thin line between plant protection and phytotoxicity, hence a more detailed study of the synthesized nanoparticles is required prior to their applications in agricultural field and presumably, it is more suitable to use Cu-tolerant cultivars.

Whereas Cu was toxic toward the *C. medinalis* larvae at 80 μM, Fe did not affect larval FW, RGR or development at concentrations of 65 mg/L (equivalent to 197 μM) or 265 mg/L (equivalent to 742 μM) compared to larvae feeding on control plants (Figure 1). To explain these observations, we must separately consider the effects of each Fe treatment on intracellular Fe levels and the growth of rice plants. The 65 mg/L Fe treatment increased Fe levels in the shoots to 1517 mg/kg DW at 15 DAT, which exceeds the 700 mg/kg DW critical toxicity threshold in rice, but even so, the vegetative growth of rice plants was temporarily improved under these conditions (Appendix A). Similarly, optimal vegetative growth was observed for rice plants in nutrient solutions containing 10 or 50 mg/L Fe, whereas growth inhibition due to Fe toxicity was observed at 250 and 500 mg/L Fe [35]. In contrast to the vegetative growth performance, prolonged exposure to 65 mg/L Fe until the reproductive stage reduced the thousand grain weight by 3.3%, even though we observed no changes in the growth of vegetative tissues (Appendix A, Table 1). The accumulation of Fe in rice shoots may have been insufficient to hamper the growth of *C. medinalis* larvae, or the robust Fe metabolism in these insects may have prevented Fe accumulation through the coordinated control of Fe absorption, transport, storage and homeostasis [36,37].

The *C. medinalis* larvae were also unaffected by the highest Fe concentration (265 mg/L) because this treatment surprisingly did not alter the intracellular Fe concentration in the shoots (Figure 1 and Appendix A). This can be attributed to a strategy I (exclusion/avoidance) mechanism deployed by rice plants to exclude soluble Fe^2+^ at the root level [38,39]. Rice plants achieve this by releasing oxygen and/or expressing enzymes that promote Fe^2+^ oxidation, leading to the formation of ferric oxide precipitates on the root surface and preventing the uptake of excess Fe^2+^ [39]. Although this mechanism has proven effective for vegetative rice growing in Fe-contaminated environments, it is less effective under prolonged exposure that extends to later growth stages because the oxidizing capacity of roots declines with age [40]. Accordingly, we observed the severe impairment of root growth and significant yield losses in the 265 mg/L Fe treatment group at the reproductive stage (Appendix A, Table 1). Given that Fe was unable to inhibit *C. medinalis* growth or development but still negatively affected rice growth and yield, it is clearly unsuitable as the basis for elemental defense against *C. medinalis* (Figure 5).

## 4. Materials and Methods

### 4.1. Plant Materials

The popular Taiwanese rice variety Taoyuan No. 3 was chosen for this study because it is highly susceptible to insect pests [41]. Seeds were surface sterilized with 1.25% NaOCl for 60 min, washed three times in distilled water, soaked in distilled water at 37 °C for 24 h and germinated on water-moistened filter paper in Petri dishes in the dark at 37 °C for 48 h. Germinated seeds of uniform size were transferred to 1 L polyethylene pots containing 700 mL sterile vermiculite soaked with 1× Kimura B nutrient solution (pH 5.0), comprising 0.36 mM (NH_4_)_2_SO_4_, 0.18 mM KNO_3_, 0.55 mM MgSO_4_, 0.18 mM KH_2_PO_4_, 61.20 µM Fe-citrate, 0.37 mM Ca(NO_3_)_2_, 2.51 µM H_3_BO_3_, 0.20 µM MnSO_4_, 0.20 µM ZnSO_4_, 0.05 µM CuSO_4_ and 0.05 µM H_2_MoO_4_ [7,42]. Each pot contained four germinated seeds. Plants were grown in a growth chamber set at 30/25 °C (day/night) with a 12 h photoperiod. The nutrient solution was renewed every other day.

Fourteen days after sowing, each pot was trimmed down to two seedlings with uniform leaf stages and the Cu or Fe treatments were applied for 16 days in different batches of experiments. In the Cu experiment, seedlings were exposed to 0.05 µM CuSO_4_ (control), or an additional 20 or 80 µM CuSO_4_. In the Fe experiment, seedlings were exposed to 15 mg/L Fe-citrate (control, equivalent to 61.20 µM Fe), or an additional 50 or 250 mg/L FeNa-EDTA (equivalent to an additional 136 or 681 µM Fe). The growth, elemental composition and phytohormone profiles of vegetative rice plants were examined at 15 DAT, whereas the growth and yield of reproductive rice plants were examined at 120 DAT. The nutrient solutions were renewed every other day. Plants were used for insect experiments 30 days after sowing.

### 4.2. Insect Experiments

We used a *C. medinalis* colony originally collected from the Taichung District Agricultural Research and Extension Station, COA, Changhua, Taiwan. *C. medinalis* larvae were reared on White Pearl maize seedlings (Known-You Seed Co., Kaohsiung City, Taiwan) according to a modified maize seedling rearing method [43] and moths were fed on a 10% (*w*/*v*) sucrose solution. The insects were kept inside BugDorm-4 mesh cages (MegaView, Taiwan) in a growth chamber set at 30/25 °C (day/night) with 55 ± 5% relative humidity and a 12 h photoperiod.

Third-instar *C. medinalis* larvae were weighed and then placed on the newly expanded leaves of 30-day-old rice plants, one larva per plant. Each experimental group consisted of 16 insects. A plastic cover with mesh cloth was placed over each pot to prevent larvae from escaping. Larval FW was recorded again at 3 and 6 dpi, whereas the developmental stage (instar or pupa) was recorded at 6 and 9 dpi. The RGR of the larvae was calculated using Equations (1) and (2) [44,45]:(1)$$\mathrm{RGR}\;\mathrm{at}\;3\;\mathrm{dpi}=\frac{\;\mathrm{Weight}\;\left(3\mathrm{dpi}\right)-\;\mathrm{Weight}\;\left(0\mathrm{dpi}\right)}{\left[\left(\;\mathrm{Weight}\;\left(3\mathrm{dpi}\right)+\;\mathrm{Weight}\;\left(0\mathrm{dpi}\right)\right)÷2\right]}\;÷\;3\;\mathrm{days}$$
(2)$$\mathrm{RGR}\;\mathrm{at}\;6\;\mathrm{dpi}=\frac{\;\mathrm{Weight}\;\left(6\mathrm{dpi}\right)-\;\mathrm{Weight}\;\left(0\mathrm{dpi}\right)}{\left[\left(\;\mathrm{Weight}\;\left(6\mathrm{dpi}\right)+\;\mathrm{Weight}\;\left(0\mathrm{dpi}\right)\right)÷2\right]}\;÷\;6\;\mathrm{days}$$

### 4.3. Rice Physiological Characteristics

We measured the shoot height of 10 plants, the FW and DW of leaf blade, sheath and root tissues in four plants, and the relative chlorophyll content of 10 plants at 15 DAT. The relative chlorophyll content of the leaves was measured using a SPAD 502 Plus chlorophyll meter (Spectrum Technologies, Aurora, IL, USA) and values were calculated based on the mean of three different points on the youngest leaf.

To assess the effects of prolonged treatment at the reproductive stage, rice plants were exposed to 0.05 (control) or an additional 10, 20 or 30 µM CuSO_4_ (Cu experiments), or to 15 mg/L Fe-citrate (control) or an additional 50 or 250 mg/L FeNa-EDTA (Fe experiments) for 120 days, beginning 14 days after sowing. We measured shoot height (eight plants), relative chlorophyll content (four plants), days to booting and heading (eight plants) and yield components (eight plants). The number of days to heading was estimated as previously described [46]. The grain yield (in grams per plant) for potted plants was calculated using Equation (3):(3)$$\mathrm{Grain}\;\mathrm{yield}\;(g/\mathrm{plant})=\frac{\;\mathrm{Panicle}\;\mathrm{number}\;}{\;\mathrm{Plant}\;}\times \frac{\;\mathrm{Grain}\;\mathrm{number}\;}{\;\mathrm{Panicle}\;}\times \frac{\;\mathrm{Total}\;\mathrm{grain}\;\mathrm{weight}\;}{\;\mathrm{Grain}\;\mathrm{number}\;}$$

### 4.4. Elemental Analysis

Root and shoot tissues from vegetative rice plants at 15 DAT were harvested separately for elemental analysis as previously described [47]. The tissues were washed in ice-cold 10 mM CaCl_2_ and again in Millipore water before drying at 70 °C for 3 days. The dried tissues were cut into fine pieces. Shoot sample (ca. 50 mg) and root sample (ca. 20 mg) from each plant were transferred into a Teflon vessel and digested with 5 mL 69% HNO_3_ (Suprapur, Merck, Kenilworth, NJ, USA) and 2 mL 37% H_2_O_2_ (Suprapur, Merck, Kenilworth, NJ, USA) before multi-element analysis by inductively-coupled plasma-optical emission spectrometry (ICP-OES) (PerkinElmer Optima 5300, Waltham, MA, USA). Tomato leaves (SRM-1573a) from the National Institute of Standards and Technology were used as a reference, giving a recovery of >72% for Cu, Fe, Zn and Mn. Elemental concentrations (mg/kg DW) in rice tissues are presented as means ± standard deviations (SD) based on four biological replicates.

### 4.5. Phytohormone Analysis

Leaf blade, sheath and root tissues from vegetative rice plants at 15 DAT were harvested separately for phytohormone analysis by LC-MS as previously described [2]. The concentrations of SA, JA and JA-isoleucine (ng/g FW) are presented as means ± SD based on three biological replicates.

### 4.6. Statistical Analysis

Statistical analysis was carried out using the lsmeans and multcomp packages in R [48,49]. Statistically significant differences (*p* < 0.05) were determined using one-way or two-way analysis of variance (ANOVA) followed by Tukey’s post hoc test as appropriate.

## 5. Conclusions

The use of micronutrient supplements to confer elemental defense against pests and pathogens in staple crops such as rice has not been investigated in detail. We found that Fe is unsuitable for this purpose because even high concentrations (265 mg/L Fe) had no effect against *C. medinalis* larvae. In contrast, 80 µM Cu inhibited *C. medinalis* growth and development but this concentration was also toxic to the rice plants. Even so, Cu-induced elemental defense in rice may be possible if the efficacy of the pesticide on the leaf surface can be increased without affecting intracellular Cu levels, for example, by the topical application of Cu nanoparticles.

## Figures and Tables

**Figure 1 plants-11-01104-f001:**
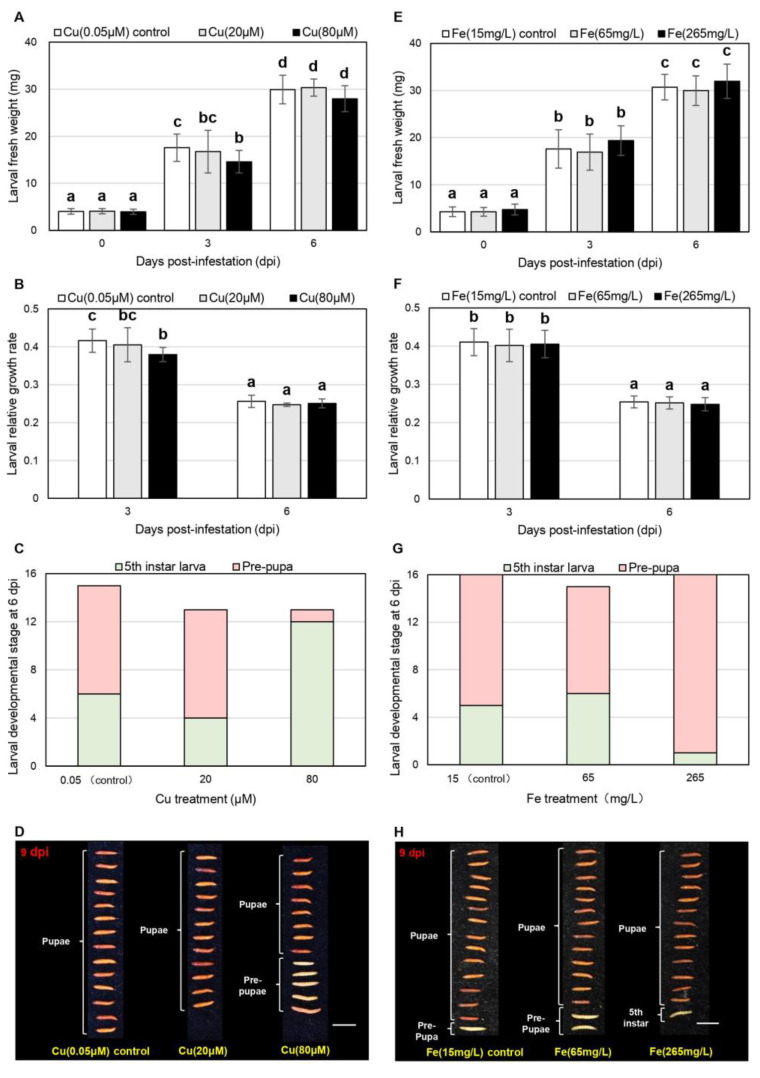
Effects of Cu and Fe treatments on *C. medinalis* growth and development at 3, 6 and 9 days post-infestation (dpi): (**A**) Larval fresh weight (FW) at 0, 3 and 6 dpi; (**B**) larval relative growth rate (RGR) at 3 and 6 dpi; (**C**,**D**) developmental stages of *C. medinalis* at (**C**) 6 dpi and (**D**) 9 dpi (scale bar = 1 cm) under Cu treatments; (**E**) Larval FW at 0, 3 and 6 dpi; (**F**) larval RGR at 3 and 6 dpi; developmental stages of *C. medinalis* at (**G**) 6 dpi and (**H**) 9 dpi (scale bar = 1 cm) under Fe treatments. Data are means ± SD (larval FW, *n* = 12–16; larval RGR, *n* = 9–16; developmental stages, *n* = 12–16). Two-way ANOVA and Tukey’s post hoc test were used for FW and RGR measurements, with different letters denoting a significant difference (*p* < 0.05).

**Figure 2 plants-11-01104-f002:**
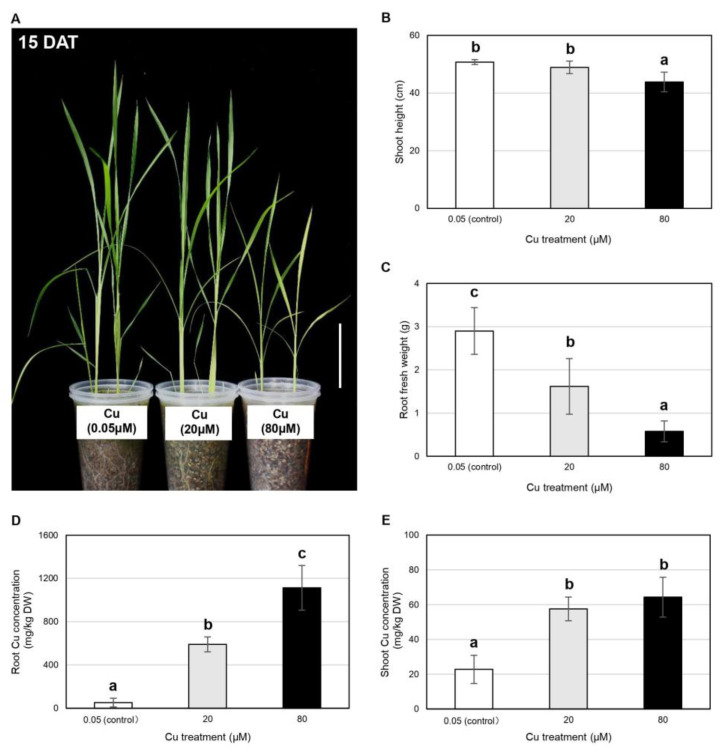
Effect of 20 and 80 µM Cu on the physiology of vegetative rice plants at 15 DAT: (**A**) Shoot morphology (scale bar = 10 cm); (**B**) shoot height; (**C**) root fresh weight (FW); (**D**) root Cu concentration (mg/Kg DW) and (**E**) shoot Cu concentration (mg/Kg DW) are shown. Data are means ± SD (shoot height, *n* = 10; root FW, *n* = 4; tissue Cu concentrations, *n* = 4). One-way ANOVA and Tukey’s post hoc test were used for each measurement, with different letters denoting a significant difference (*p* < 0.05).

**Figure 3 plants-11-01104-f003:**
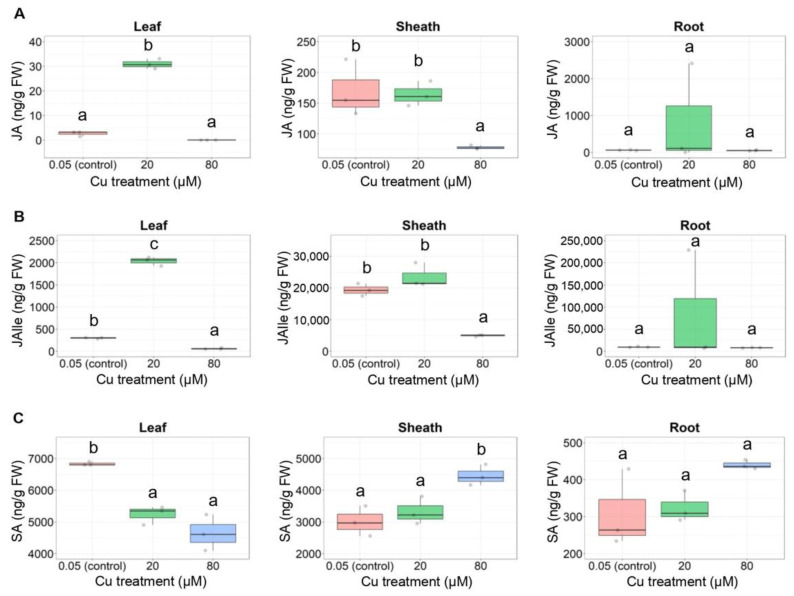
Effect of 20 and 80 µM Cu on defense-related phytohormone concentrations in rice tissues at 15 DAT: (**A**) Jasmonic acid (JA); (**B**) JA-isoleucine (JAIle) and (**C**) salicylic acid (SA) concentrations are shown in three tissues. Data are means ± SD (*n* = 3). One-way ANOVA and Tukey’s post hoc test were used for phytohormone measurements in each tissue, with different letters denoting a significant difference (*p* < 0.05).

**Figure 4 plants-11-01104-f004:**
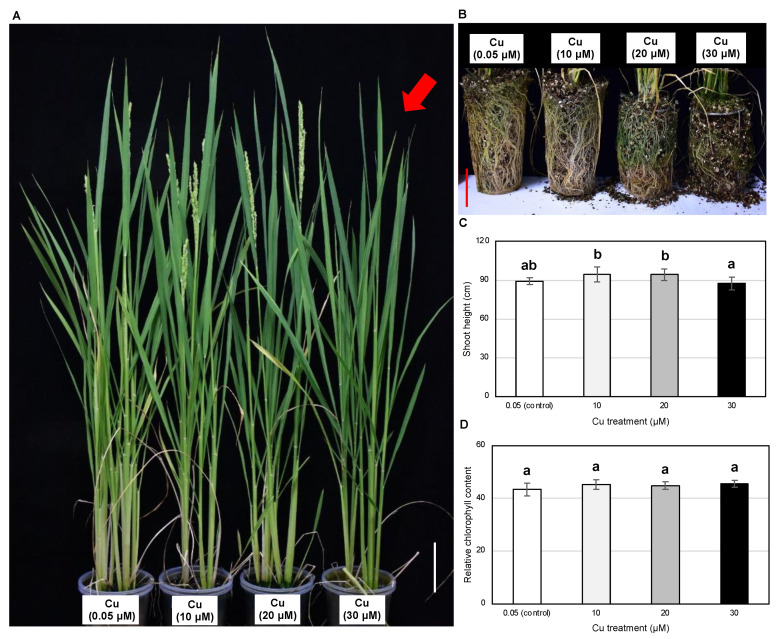
Effects of 10, 20 and 30 µM Cu on the physiology of rice plants at the reproductive stage: (**A**) Shoot morphology during heading (scale bar = 10 cm); (**B**) root morphology (scale bar = 5 cm); (**C**) shoot height and (**D**) relative chlorophyll content (SPAD) are shown. Data are means ± SD (shoot height, *n* = 8; relative chlorophyll content, *n* = 4). Red arrow indicates delayed heading in a plant exposed to 30 µM Cu. One-way ANOVA and Tukey’s post hoc test were used, with different letters denoting a significant difference (*p* < 0.05).

**Figure 5 plants-11-01104-f005:**
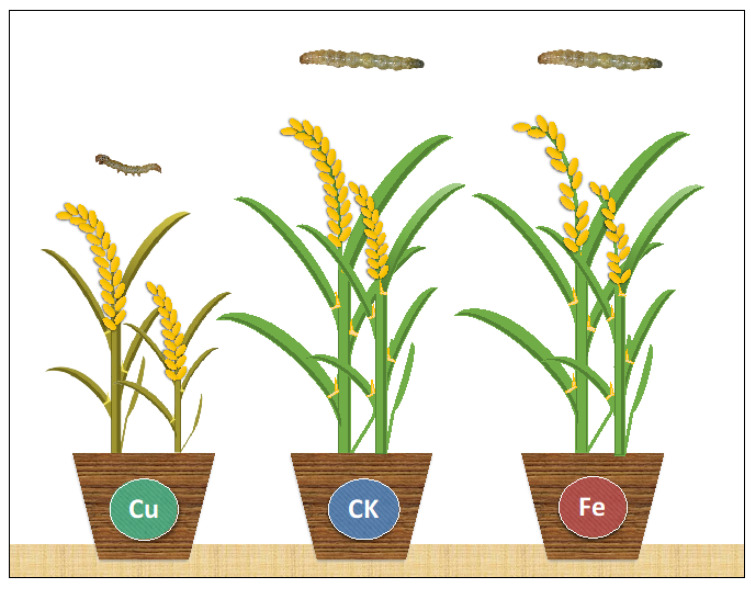
Summary of the elemental defense potential of Cu and Fe against the *C. medinalis*. High levels of Cu impede the growth and development of *C. medinalis* but the elemental defense potential is limited because this treatment seriously affects the growth and development of rice plants at the vegetative and reproductive stages. High levels of Fe offer no elemental defense potential because the treatment does not affect the growth of *C. medinalis* but does inhibit root growth and reduce rice yields at the reproductive stage.

**Table 1 plants-11-01104-t001:** Effect of Cu and Fe treatments on the yield components of rice. Grain yield is the product of panicle number per plant, grain number per panicle and the thousand grain weight/1000. For each yield component, the data are means ± SD (*n* = 8), with significance determined by one-way ANOVA followed by Tukey’s post hoc test (*p* < 0.05, as denoted by different letters).

Treatment	Concentration	Panicle Number per Plant	Grain Number per Panicle	Thousand Grain Weight (g)	Grain Yield (g/Plant)
Cu	0.05 μM (control)	2.75 ± 0.46 a	75.14 ± 23.61 a	28.48 ± 0.34 c	5.89 ± 1.53 a
	10 μM	2.63 ± 0.52 a	65.90 ± 24.94 a	29.49 ± 0.44 d	5.10 ± 1.40 a
	20 μM	2.13 ± 0.35 a	81.18 ± 25.33 a	27.44 ± 0.12 b	4.73 ± 0.77 a
	30 μM	2.25 ± 0.71 a	76.20 ± 21.70 a	26.39 ± 0.21 a	4.76 ± 0.82 a
Fe	15 mg/L (control)	3.00 ± 0.53 b	78.88 ± 26.71 a	27.62 ± 0.15 b	6.49 ± 0.40 b
	65 mg/L	2.50 ± 0.53 ab	81.65 ± 30.52 a	26.72 ± 0.26 a	5.45 ± 0.93 ab
	265 mg/L	2.25 ± 0.46 a	75.33 ± 29.09 a	27.01 ± 0.24 a	4.58 ± 1.23 a

## Data Availability

Data is contained within the article and Appendix A.

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
