# Peer review of "Exogenous Copper Application for the Elemental Defense of Rice Plants against Rice Leaffolder (Cnaphalocrocis medinalis)"

_plants, 2022, doi:10.3390/plants11091104_

Round 1

Reviewer 1 Report

This is a nice study that will provide an interesting addition to the literature on the impact of metals on rice plant growth and the development of an important rice pest.  I had no concerns with the general experimental design or the statistical analyses.  The inclusion of plant chemical analyses (both of defensive compounds and metal concentrations) added considerably to the interpretations of the data.  This was nicely done.

I did have two relatively minor concerns. 

  1. The authors repeatedly stated (in the abstract, results section, and conclusions) that there is considerable potential for the use of Cu-based nanoparticles in pest management on rice because this could achieve higher concentrations of Cu on the leaf surface while preventing absorption by the plant.However, this was not tested, nor was any reference presented that indicated 1) the plant would not accumulate Cu if Cu nanoparticles were topically applied, 2) levels of Cu nanoparticles would deliver high enough concentrations to impact pest growth and damage or, 3) the presence of Cu-based nanoparticles would not otherwise effect plant growth and development.  Thus, I believe this statement is too speculative for inclusion in the abstract and should probably only be included as a suggestion for possible future research in the conclusions section.

  1. The information on the ICP-OES analysis is not adequate. In order for this to be repeated (a basic tenant of science), readers will need to know what machine was used, and what analytical conditions were used. No references were cited that could provide this information. Also, how many plants were tested and how were the locations on the plants standardized (if at all). In addition, what was the recovery rate?  ICP-OES analyses of plant tissues generally include spiked samples of a reference plant tissue to show the recovery rate from a plant matrix.  Simple percent recovery rates from a known concentration of a sample metal alone are not as useful but would still provide useful information. If possible, this information should be provided.

Author Response

Dear Editor and Reviewers,

Please find enclosed our revised manuscript (Manuscript ID: plants-1683409) entitled “Exogenous copper application for the elemental defense of rice plants” as a research article in Plants. For your information, we resubmitted two versions of revised manuscript documents, one is the final version in PDF and the other version in WORD shows the revised parts using “Track Changes”. The files can be downloaded from the link. https://www.space.ntu.edu.tw/navigate/s/0F0D7D3572C7405DAFDB6E72AAD668BAQQY

We wish to thank the editor, and all reviewers for the meticulous review, thoughtful and constructive feedback on this manuscript. The manuscript has been revised according to the comments and suggestions that we believe have increased the scientific value of the manuscript significantly. Our responses to all the comments are as detailed in the attached “Response to Reviewers’ Comments” file. Please see the attachment.

Yours sincerely,

Dr. Ya-Fen Lin

Department of Agronomy

National Taiwan University

Taipei, Taiwan

Reviewer 2 Report

The manuscript “ Exogenous copper application for the elemental defense of rice plants against rice leaffolder (Cnaphalocrocis medinalis)” provided findings for insect and plant exposure to sublethal concentrations of 2 metals. The study was well designed and answered questions related to the concentration range of Fe and Cu treatments that would effectively control a rice pest in the larval stage, yet not cause a negative effect to plant growth and grain yield. The manuscript could be accepted with minor revisions.

Results

Figure 1. A, B, C, E, F, G – the axis labels and legend font size is small and difficult to read, please increase size.

Could statistics be performed on the Figure 1 C and G larval development stage?

Discussion

Lines 249-252 - Is there any evidence that nanoparticles with Cu or other metals would not cause toxicity to plants (rice)? Please provide references for the application of nanoparticles in insect pest control to support this suggestion.

Could the authors elaborate on whether a rice cultivar (other than the one in this study) more tolerant to metals (Cu) would be more suitable for this type of application – was the cultivar chosen too susceptible to Cu?

Figure 5 – Suggest using actual images of the C. medinalis larvae that represent the larger (CK and Fe plants) and smaller (Cu plants) larvae rather than the images provided (or use a more realistic cartoon version of a larvae).

Author Response

(The authors gave the same response as above.)
